# Physiological Responses of a Grapefruit Orchard to Irrigation with Desalinated Seawater

**DOI:** 10.3390/plants13060781

**Published:** 2024-03-09

**Authors:** Josefa M. Navarro, Alberto Imbernón-Mulero, Juan M. Robles, Francisco M. Hernández-Ballester, Vera Antolinos, Belén Gallego-Elvira, José F. Maestre-Valero

**Affiliations:** 1Irrigation and Stress Physiology Group, Department of Bioeconomy, Water and Environment, Murcia Institute of Agri-Food Research and Development (IMIDA), c/Mayor s/n, 30150 Murcia, Spain; josefam.navarro2@carm.es (J.M.N.); juanm.robles@carm.es (J.M.R.); franciscom.hernandez4@carm.es (F.M.H.-B.); verantolinos@gmail.com (V.A.); 2Agricultural Engineering Center, Technical University of Cartagena, Paseo Alfonso XIII 48, 30203 Cartagena, Spain; belen.gallego@upct.es (B.G.-E.); josef.maestre@upct.es (J.F.M.-V.)

**Keywords:** boron, chloride, citrus, leaf toxicity, sodium, water management

## Abstract

Desalinated seawater (DSW) has emerged as a promising solution for irrigation in regions facing water scarcity. However, adopting DSW may impact the existing cultivation model, given the presence of potentially harmful elements, among other factors. A three-year experiment was carried out to assess the short-term effects of four irrigation waters—freshwater (FW), DSW, a mix 1:1 of FW and DSW (MW), and DSW with low boron (B) concentration (DSW–B)—on a ‘Rio Red’ grapefruit orchard. These irrigation waters exhibited varying levels of phytotoxic elements, some potentially harmful to citrus trees. Sodium (Na^+^) and chloride (Cl^−^) concentrations exceeded citrus thresholds in all treatments, except in DSW−B, whilst B exceeded toxicity levels in DSW and MW treatments. Leaf concentrations of Cl^−^ and Na^+^ remained low in all treatments, whereas B approached toxic levels only in DSW and MW–irrigated trees. The rapid growth of the trees, preventing excessive accumulation through a dilution effect, protected the plants from significant impacts on nutrition and physiology, such as gas exchange and chlorophyll levels, due to phytotoxic elements accumulation. Minor reductions in photosynthesis in DSW–irrigated trees were attributed to high B in leaves, since Cl^−^ and Na^+^ remained below toxic levels. The accelerated tree growth effectively prevented the substantial accumulation of phytotoxic elements, thereby limiting adverse effects on tree development and yield. When the maturation of trees reaches maximal growth, the potential accumulation of phytotoxic elements is expected to increase, potentially influencing tree behavior differently. Further study until the trees reach maturity is imperative for comprehensive understanding of the long-term effects of desalinated seawater irrigation.

## 1. Introduction

In recent years, water scarcity has become a major concern for farmers in arid and semi-arid regions [1]. Owing to widespread climate change, the reduction in freshwater resources is threatening the sustainability of irrigated agriculture, and hence food security [2,3]. Furthermore, the upcoming scenario of exorbitant food demand, added to the intensification of water deficits, requires a new agricultural management model to be found, based on alternative water resources to encourage current agri-food production in water-starved areas [2,4]. The Mediterranean area suffers from a severe water scarcity and it is one of the most water-stressed areas in Europe [4,5,6], yet it is one of the regions with the highest fruit and vegetables production and exportation rates worldwide. This has been achievable due to its excellent edaphoclimatic conditions and very competitive agriculture strategies, which allow for extremely efficient use of the available water supplies [2]. Given this scenario, the use of unconventional water resources such as reclaimed water and desalinated seawater (DSW) is today considered as a promising stimulus for irrigation in this area [7,8]. However, its use for irrigation must be undertaken with caution, since it may condition the current agriculture framework [1,2]. One of the main drawbacks of its use lies in its chemical composition, as DSW is characterized as having a very low concentration of essential nutrients (Ca^2+^, Mg^2+^, or SO_4_^2−^), albeit with a high concentration of B, Na^+^, and Cl^−^ [7,9]. Proper management is therefore required since most woody crops are rather sensitive to B, Cl^−^, and Na^+^ leaf toxicity [10].

The southeast of Spain is a major citrus producer, due to its warm Mediterranean climate and fertile soil that create an ideal environment for citrus cultivation [11,12]. Because of the high structural deficit of water in southeastern Spain, the desalination of seawater has been promoted politically to sustain socioeconomic progress and food production in this important citrus-producing area [9]. However, using DSW for citrus irrigation requires caution due to its high salt content, since particularly citrus have been shown to be vulnerable to soil salinity and extremely sensitive to the concentration of B [10]; waters with a B concentration above 0.5 mgL^−1^ can induce toxicity problems [9,13]. Common toxic effects of a high B concentration in citrus trees are leaf burn, yellowing, and browning, fruit with sunken areas, damaged root systems impacting water and nutrient uptake, and premature flower and fruit drop, leading to reduced yield [14]. Moreover, many studies have shown that citrus trees are sensitive to high levels of Cl^−^ and/or Na^+^ in soil and water [15,16], so the excessive salt amounts in DSW could damage plant roots, thereby hindering nutrient absorption and tree growth. In principle, when citrus are irrigated with high salts concentrations, the soil and leaf accumulations may lead to damage in the tree by reducing the net assimilation of CO_2_; they can also damage leaves and roots, disrupt nutrient absorption, and cause water stress, ultimately influencing tree growth, development, and above all, productivity [16,17,18]. Overall, their ability to control these environmental stresses seems to depend on the rootstock [18,19]. Recent research conducted under controlled conditions has shown that the rootstock can modulate the citrus plant response to DSW irrigation [20,21].

When referring specifically to physiological and nutritional effects, the studies available in the literature to date have mainly evaluated the detrimental effects of using DSW for citrus irrigation under controlled conditions. The research showed that high soil salinity and the accumulation of B, Cl^−^, and Na^+^ in citrus trees can impede the uptake of certain nutrients due to chemical competition and produce specific toxicity problems in the plants [20,21]. When DSW was used to irrigate a mandarin orchard, non-significant toxic responses (leaf toxic ions accumulation, low plant water relations, or low crop gas exchange) were observed; this was likely due to the young age of the trees [3]. On the contrary, other studies in lemon trees irrigated with DSW for three consecutive years showed reductions in stomatal conductance and net photosynthesis, along with a noticeable accumulation of B in leaves [22]. In any case, the variety of factors that seem to affect the crops’ response to DSW irrigation can hinder the establishment of sound conclusions; further information regarding its primary effects is therefore still needed.

Sustainable water management and balancing plant needs are clearly crucial for agricultural and environmental health. Therefore, the present study aims to initiate a long-term experiment to provide guidance on the effects of DSW utilization in semi-arid areas where low-quality waters are commonly used for irrigation, particularly for crops potentially sensitive to DSW, such as citrus. Our hypothesis is that the irrigation of grapefruits with DSW could alter the physiological response of the tree compared to conventional irrigation due to the accumulation of phytotoxic elements, particularly boron, in the soil and plant. The main objective of this experiment was to assess, for the first time, the main physiological implications of irrigating a ‘Rio Red’ grapefruit orchard with DSW. Therefore, the effects of the accumulation of phytotoxic elements on the soil and the plant due to the irrigation with DSW for three consecutive seasons were evaluated, as well as their implications on the gas exchange parameters, tree water relations, vegetative growth, and fruit yield. The significance of this study lies in its contribution to sustainable water management practices, offering crucial insights into the utilization of DSW in semi-arid regions for irrigation, particularly for sensitive crops like citrus. Grapefruit was selected due to its relatively limited research attention, although it holds great importance due to its relevant production in Europe and the Mediterranean area. In addition, it is known for its susceptibility to excessive B levels. Two previously described agronomical practices [9] were assessed in this study in order to determine the potential risk of crop damage associated with the high concentration of B in DSW. These practices aim to diminish the B concentration in DSW, either by blending DSW and water with a low B content, or by reducing its concentration with an on-farm reverse osmosis (RO) system [23].

## 2. Results and Discussion

### 2.1. Irrigation Water Quality

The water quality of the different water sources used during the experimental trial showed notable differences (Table 1 and Figure 1). After a one-stage RO process at the coastal desalination plant to produce the DSW, a large removal of salts occurred and the EC of this water fell to 0.89 dS m^−1^ (Table 1). This value was significantly lower than that registered by the water provided by the Campo de Cartagena Irrigators Community (FW), which presented the highest EC of the experiment. On the contrary, the DSW–B treatment had the lowest value since a second on-farm RO stage was conducted.

One of the main issues about the quality of DSW concerns the lack of essential nutrients for the plants, such as Ca^2+^, Mg^2+^, and SO_4_^2−^ (Table 1). Desalination through RO processes not only separates the undesirable salts from the water, but also removes minerals that are essential nutrients for plant growth [7]. Therefore, DSW significantly reduced the concentration of nutrients with regard to FW (Table 1), and MW presented intermediate concentrations, as would be expected after mixing both sources of water at 50%. Moreover, DSW–B received two RO processes (coastal and on-farm), and the final concentrations of nutrients were much lower than those of the rest of the treatments. However, to guarantee agricultural needs, the missing essential nutrients (Ca^2+^, Mg^2+^, and SO_4_^2−^) were implemented in all treatments through a fertilization program similar to that applied by local farmers to citrus orchards.

The main objective of the RO process is the elimination of undesirable salts in seawater, mainly Na^+^ and Cl^−^, both ions represent the main source of salt in seawater (10.8 g L^−1^ and 19.5 g L^−1^, respectively) [24]. However, in addition to the high percentage of Na^+^ and Cl^−^ retained by the RO membranes (99.8 and 98.9%), respectively [25], after the desalination process, the final concentrations of these ions still remained high (Figure 1). In this experiment, DSW contained the highest Na^+^ and Cl^−^ concentrations of all the treatments in the first half of the experiment, whilst MW had values between those found in DSW and in FW. Additionally, DSW–B presented the lowest Na^+^ and Cl^−^ concentrations after the removal of salts produced by the second RO process (Figure 1). The average concentration of Na^+^ measured during the experimental period in all treatments, except in DSW–B, exceeded the maximum thresholds established for citrus irrigation (115 mg L^−1^) [26] and hence, might cause detrimental effects on crops. With regard to the Cl^−^ concentrations, except for the DSW–B treatment, all the treatments presented concentrations above the 152 mg L^−1^ established as the threshold for citrus by some authors [27], but below the 238 mg L^−1^ [28] and 350 mg L^−1^ thresholds proposed by others [29].

**Figure 1 plants-13-00781-f001:**
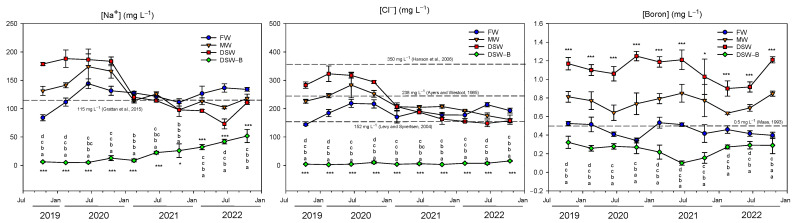
Concentrations of the phytotoxic elements, Na^+^, Cl^−^, and B in the water resources used during the experimental work (FW: fresh water; MW: mixed water; DSW: desalinated seawater; and DSW–B: DSW with reduced boron). The dashed line in the figure determines the phytotoxicity threshold suggested for Na^+^ [26], Cl^−^ [27,28,29], and B [16]. Data were taken between July 2019 and December 2022. Each point represents the average of four samples. * *p* < 0.05; *** *p* < 0.001; ns: not significant. For each date, different letters indicate significant differences according to Duncan’s multiple range test at the 95% confidence level.

Conversely, as RO membranes are poorly selective for B (50–80% at normal feedwater pH values) [30], B concentrations were high after the desalination process (Figure 1). In this experiment, DSW presented the highest B concentrations throughout the experiment (1.1 mg L^−1^ on average), whilst the concentration in FW was significantly lower (0.45 mg L^−1^). This could lead to phytotoxicity and detrimental effects on crop productivity if DSW were used as the sole source of irrigation [31]. The B concentration in MW (0.76 mg L^−1^) was between those recorded for both DSW and FW, whereas the value registered for DSW–B was the lowest one (0.24 mg L^−1^). Thus, the average B concentration measured during the experimental period in DSW and MW exceeded the maximum threshold of 0.5 mg L^−1^ referenced for citrus irrigation [16], whereas in FW it was very close to that threshold and in DSW–B it was far below it (Figure 1). These concentrations were found to be within the range reported in previous studies [3,22,32]. In fact, recent studies [20,21,22] have detected toxic B levels in the leaf and a decline in physiological parameters in lemon trees with similar DSW water quality properties. In this sense, the second RO stage performed on the farm was able to decrease the overall DSW concentration of B below the phytotoxic threshold (Figure 1), thereby avoiding possible detrimental effects on the tree.

### 2.2. Evolution of Toxic Elements in the Soil

The sodium concentration in the soil ranged from 100 to 200 mg L^−1^ at the beginning of the experiment (Figure 2) and fluctuated throughout the experiment. After three years of irrigation, its final concentration in soil was 2–3 times higher (280–460 mg L^−1^). Soils irrigated with DSW had the highest Na^+^ concentration overall (average of 365 ± 36 mg L^−1^), whereas those irrigated with DSW–B had the lowest (average of 128 ± 20 mg L^−1^). The soils irrigated with FW presented an intermediate Na^+^ concentration (213 ± 20 mg L^−1^), whilst those irrigated with MW had slightly higher Na^+^ levels (245 ± 26 mg L^−1^). These Na^+^ values in the soil were significantly correlated with the different Na^+^ concentrations of the irrigation waters used in the four treatments (Table 2).

Similarly, the Cl^−^ concentrations at the beginning of the experiment were between 50 and 150 mg L^−1^, rising to 200–450 mg L^−1^ at the end of the experiment, i.e., increasing by 2–5 times (Figure 2). However, unlike with the Na^+^ accumulation in the soil, the highest Cl^−^ accumulation was not found in soils irrigated with DSW, since they presented similar Cl^−^ concentrations to those of the FW and MW treatments. Only soils irrigated with DSW–B had soil Cl^−^ concentrations below the lower limit of the range 178–355 mg L^−1^ beyond which many sensitive plants are injured [29], whereas in the rest of treatments the concentrations were almost always below the higher limit. Although only the soils of the DSW–B treatment had significantly lower Cl^−^ concentrations than the rest of treatments throughout the experiment, a significant correlation was found between the Cl^−^ levels of the water and the Cl^−^ accumulation in the soil (Table 2).

With regard to B, its soil concentration fluctuated throughout the experiment, but remained almost constant at the final measurements (Figure 2). In fact, the soil B concentration increased slightly, from 1.2–1.3 mg kg^−1^ soil at the beginning of the experimental period to 1.0–2.3 mg kg^−1^ soil at the end, increasing by only 1.7 times in the soils from the DSW treatment. Those soils irrigated with DSW had the highest B concentration due to its higher concentration in the water source. Recent studies using DSW for citrus irrigation also found important accumulations of B in soils irrigated with DSW [21], albeit without the B content exceeding medium–high levels for agricultural soils (0.21–1.10 mg kg^−1^ soil) [34]. Consequently, the lower the proportion of DSW in the irrigation blend, the lower the concentration of B in the soil (DSW > MW > FW > DSW–B; Figure 2). In this sense, a strong positive correlation was found between the B concentrations of the water used in the treatments of this experiment and the soil B concentration (Table 2). Some of the B levels found in the experiment could prove toxic for citrus. The range between B deficiency and toxicity is extremely narrow compared to other elements [35] and the B concentrations found in all of the treatments of this experiment were between 0.9 and 2.3 mg kg^−1^, the vast majority of these values being between 1.2 and 3.0 mg kg^−1^ soil; such a range of concentration in soil has been deemed to be very high [34]. Citrus trees are considered to be very sensitive to B toxicity [26] and some authors have proposed 1.5 mg kg^−1^ soil of hot-water-extractable B as the threshold for sensitive plant species [33]. In fact, only soils irrigated with DSW–B (average of 1.2 mg kg^−1^ soil) or FW (average of 1.3 mg kg^−1^ soil) had B concentrations below the threshold of 1.5 mg kg^−1^ soil, whilst the remaining treatments (1.5 mg kg^−1^ soil and 1.8 mg kg^−1^ soil for MW and DSW, respectively) had B concentrations above this level, and could thus present B toxicity problems.

### 2.3. The Accumulation and Partitioning of Phytotoxic Elements in the Trees

The trees were irrigated with water of different characteristics throughout the experiment (Table 1 and Figure 1). Since the amounts of the phytotoxic elements (Na^+^, Cl^−^, and B) in the four treatments were different, their concentrations in the trees were studied. The evolution of these elements in spring bud leaves showed similar behavior throughout the three years of the experiment, so the average data of the three years were evaluated.

Despite being the oldest leaves on the tree, during the first months of the experiment the old leaves were still very young, and they did not accumulate Na^+^ in any treatment until June 2020 (almost constant concentration, 0.10–0.15%, Figure 3). Subsequently, the Na^+^ presence progressively increased up to 0.16–0.22% in October 2020. From that moment on, the Na^+^ concentrations were higher in DSW-irrigated trees than in the rest of the treatments, although significant differences were only occasionally found. The spring bud leaves accumulated slightly lower amounts of Na^+^ than older leaves and, although there were no differences with the rest of treatments, DSW-irrigated trees had the highest Na^+^ levels throughout the year (Figure 3). In any case, in both old and young leaves, concentrations remained below the maximum Na^+^ threshold of 0.25% proposed for citrus [26], even though the Na^+^ concentration in the water for all the treatments (except for DSW–B, Figure 1) exceeded the toxic threshold of 115 mg L^−1^ proposed for citrus [26]. Moreover, although the irrigation with different Na^+^ concentrations produced different Na^+^ accumulations in the soil (Figure 2 and Table 2), no significant correlations were found between the water or soil Na^+^ concentrations and the Na^+^ concentration in the leaves. Additionally, higher Na^+^ levels due to DSW irrigation were not found in the root examined in May 2021 (Figure 3). Therefore, the higher Na^+^ concentration of soils irrigated with DSW during the experiment was not enough to produce an increase in Na^+^ in the roots nor in the spring bud leaves, and only the oldest leaves occasionally had higher Na^+^ levels than in the rest of the treatments.

The higher concentration of Na^+^ in DSW could produce an imbalance of essential nutrients since direct competition between ions can reduce the nutrient uptake by plants [36]. Previous studies with DSW in citrus irrigation have shown high Na^+^/Ca^2+^, Na^+^/K^+^, and Na^+^/Mg^2+^ ratios in plant tissues when DSW was used for citrus irrigation [20]. In our experiment, since the DSW irrigation did not cause a large accumulation of Na^+^ in the leaves, non-relevant nutritional disorders were found. In this sense, the irrigation with DSW only affected the mineral nutrition of old leaves, which had the highest Na^+^/Ca^2+^ ratio (Figure 4), mainly due to the higher amount of Na^+^ taken up by the roots to the leaves. However, the Na^+^/Mg^2+^ ratio was the highest in old leaves and roots of DSW-irrigated trees (Figure 4) due to the increase in Na^+^ concentrations in leaves -. Similar findings were found in spring bud leaves but without significant differences. On the other hand, K^+^ does not always decline in citrus leaves under salinity [37]; in our experiment, no increase in the Na^+^/K^+^, ratio was found in roots or leaves due to the high Na^+^ concentrations of the DSW treatment (Figure 4).

The Cl^−^ concentration in the treatments (except in DSW–B, Figure 1) fell within the range of 152–238 mg L^−1^, proposed as a range in which Cl^−^ toxicity can occur in citrus [26]. However, these concentrations were below the threshold of 350 mg L^−1^ proposed by other authors [29]. Despite these high Cl^−^ concentrations in the irrigation water, the leaf Cl^−^ concentrations found in the trees of all treatments were well below the leaf toxic threshold of 0.6% proposed for citrus [38] (Figure 5). Although the oldest leaves accumulated the highest Cl^−^ concentrations, no clear buildups were observed in the leaves of the DSW treatment, whilst the DSW-irrigated trees had the highest Cl^−^ concentrations in the spring bud leaves. Nevertheless, a low correlation was found between the Cl^−^ of the irrigation water and the Cl^−^ found in the leaves (Table 2), since leaf Cl^−^ levels were similar in all the treatments (except the young leaves of the DSW), even in those trees irrigated with very low Cl^−^ (trees irrigated with DSW–B, Figure 1). Therefore, although the Cl^−^ accumulation in soil was significantly correlated with the Cl^−^ of the irrigation water (Table 2 and Figure 2), the leaf Cl^−^ levels were not correlated with those in the irrigation water or soil. Unlike what happened with the Na^+^ in roots, the DSW-irrigated trees had the highest Cl^−^ concentrations in roots, whilst the lowest were found in the DSW–B treatment, in accordance with the concentrations of Cl^−^ in the irrigation water of the treatments prior to root sampling in May 2021 (Figure 1). Thus, the Cl^−^ accumulated in the soil due to the irrigation with high Cl^−^ concentrations significantly increased its root concentration but without being enough to achieve a high presence in the leaves.

Although the B concentration in DSW and MW exceeded the maximum threshold of 0.5 mg L^−1^ (Figure 1) proposed for citrus [14,39], the oldest leaves were still too young to accumulate large amounts during the first months of the experiment. Therefore, until June 2020, B levels remained approximately constant between 90 and 140 mg kg^−1^ DW (Figure 6); these values were slightly higher than the threshold of 100 mg kg^−1^ DW above which B can become toxic, but still in the low end of the 100–300 mg kg^−1^ DW range, where slight to moderate damage can occur [26]. Boron concentrations in old leaves significantly increased from June to August 2020 up to concentrations close to 250 mg kg^−1^ DW, with the range of 250–260 mg kg^−1^ DW being the critical leaf B level in citrus when toxicity occurs [40]. The B concentration in spring bud leaves increased progressively throughout the year: after two months it was almost three times the spring concentration and four times their B concentration in late autumn (Figure 6). This reveals how B was quickly accumulated in the young tissues of the plant. The spring bud leaves of DSW- and MW-irrigated trees reached their highest B concentration in autumn (over 200 mg kg^−1^ DW), accumulating significantly more B than in the FW or DSW–B treatments (Figure 6). These high concentrations found in spring leaves 6–7 months after sprouting were very close to the critical range of 250–260 mg kg^−1^ DW [41]. The lowest B values in spring bud leaves were found in the trees irrigated with DSW–B, since the B concentration in that type of water was the lowest (Figure 1). Despite this, Appendix A shows images of buds, young leaves, and old leaves from the DSW treatment, with no phytotoxic effects observed in any of the cases. No direct correlation was found between B in the irrigation water and the leaves (Table 2). However, its concentration in the irrigation water was significantly correlated with its presence in the soil, and in turn with the leaf B concentration, as has been previously found in other studies using DSW for citrus irrigation [22]. This also indicates a higher B mobility in soil solution under the DSW treatment than under the other treatments. On the other hand, in our experiment, roots had much lower B than leaves, and the roots with the highest concentration were those of DSW-irrigated trees (Figure 6). Other authors have also found relatively low B levels in roots compared to those in leaves, and non-visible symptoms of toxicity developed in roots even at very high supply concentrations [14,42,43]. In our experiment, B leaf concentration increased with its presence in the soil as well as with the exposure time. In addition, and since the major concentration was observed in leaves as compared to roots, the data revealed that B was distributed in the plant predominantly by the transpiration stream, accumulating in leaf tissues, as has been established in previous studies [41,42,43,44].

### 2.4. Plant Water Status and Physiological Responses

Plant water status was periodically measured in mature leaves from May to November (2020–2022 period), and the average of the data registered during the three years of the experiment is shown in Figure 7. In terms of the threshold value in citrus trees for the midday stem water potential (Ψ_stem_, which directly reflects the plant’s water status), all treatments in this study registered values ranging between −0.6 and −1.4 MPa during the May–November period (Figure 7), which is close to data previously reported [45]. As has been described for citrus [46], the Ψ_stem_ values suffered a seasonal behavior: a progressive decrease during the spring–summer months as a response to the elevated vapor pressure deficit (VPD), reaching their lowest values in August, and a later recovery due to the decrease in ETc and to rainfall in the subsequent months. Similar responses were found in the leaf water potential (Ψ_leaf_), with slightly lower values than for Ψ_stem_ (between −0.8 and −1.9 MPa), with the lowest values being achieved in August, and a slower recovery after the summer, as was observed for leaf water potential. In general, no clear differences were found between treatments in Ψ_stem_ or Ψ_leaf_ throughout the experiment (Figure 7). The seasonal decrease in Ψ_leaf_ did not produce a similar decrease in osmotic potential (Π), and only a slight decrease was detected in Π during the months of elevated VPD. This decrease in Π was not enough to avoid changes in the turgor values, and leaf turgor suffered a similar seasonal decrease to that described for Ψ_leaf_ (Figure 7).

The slight reduction in Π throughout the year was only due to the leaf accumulation of the phytotoxic elements Na^+^, Cl^−^, and B (Figure 4, Figure 6 and Figure 7), since no contribution of organic solutes as proline or quaternary ammonium compounds (QACs) was detected in response to the seasonal osmotic potential decrease (Figure 8). In general, no differences were found in the concentration of proline or QACs throughout the experiment with the irrigation using different types of water. Despite the results reported by other authors, which determined that QACs are involved in the adaptation of citrus to B toxicity [41], we found no changes in QACs in the leaves of DSW-irrigated trees and only those trees irrigated with DSW or MW had significantly higher proline concentrations in June than those of FW or DSW–B (Figure 8).

In addition to the plant water status analysis, a similar study of the photosynthetic machinery (gas exchange, chlorophyll fluorescence, and chlorophyll concentration) was also carried out for the same period (May–November, 2020–2022). During the summer months, the strong increase in VPD due to the low air humidity and high temperature produces a typical midday stomatal closure, even in well-watered citrus trees [47]. These high VPD conditions reduce stomatal conductance and photosynthesis while simultaneously increasing plant water losses through transpiration. In this experiment, the gas exchange parameters were measured before the midday stomatal closure (between 08:00 and 10:00 GMT), and a seasonal behavior was observed (Figure 9). Stomatal conductance (*g_s_*) increased from 0.06 mol H_2_O m^−2^s^−1^ (registered in May) to 0.17 mol H_2_O m^−2^s^−1^ (August), and slowly decreased after that maximum value. The water loss due to transpiration (*E*) presented a similar behavior throughout the same period, with values ranging from 1.3 (May) to 4.1 mmol H_2_O m^−2^s^−1^ (August). However, the photosynthetic rate (*A*) did not follow the same seasonal pattern and it increased progressively, reaching values from 7.4 to 14.1 μmol CO_2_ m^−2^s^−1^ registered in May and November, respectively (Figure 9). As a consequence, the intrinsic water use efficiency (*A/g_s_*) also increased from August to November, showing the intrinsic capacity of the leaves to exacerbate the photosynthesis rate even with a reduction in the stomatal conductance (Figure 9).

On the other hand, photosynthesis is one of the main metabolic processes impaired by excessive B [41,42,43,44] due to both stomata and non-stomata limitations. In our experiment, slight differences were found in July in photosynthetic parameters due to the type of water used for irrigation (Figure 9). Since B accumulation is influenced by transpiration, the reduction in the water transport and transpiration under high external B is a mechanism of B tolerance [48]. In this experiment, there were no large reductions in transpiration, so this tolerance mechanism did not work. In general, lower *g_s_* on DSW-irrigated trees with respect to the rest of the trees was observed throughout the experiment, which produced lower *E* and *A*, although they were only significantly different in July. Previous experiments with young lemon trees under controlled conditions showed a decrease in *A* but not in *g_s_* or *E* due to the irrigation with DSW, which produced a decrease in the intrinsic water-use efficiency [20,21]. However, no decrease in *A/g_s_* was observed due to the irrigation with DSW in this experiment. Other authors have described a similar decrease in *A* when DSW was used for citrus irrigation [22], although they also found a higher decrease in *g_s_* by the accumulation of B in the leaf, which produced an increase in the intrinsic water-use efficiency.

The decline in *A* produced by irrigation with the DSW could be attributed to other internal non-stomatal factors related to Cl^−^, Na^+^, or B toxicity, such as the reduction in the chlorophyll concentration, alterations in the carboxylation efficiency, reduced activity of photosynthetic enzymes, or alterations in the photochemical efficiency of photosystem II [20,21]. Although an excess of B in citrus trees can result in alterations of photosynthetic pigments [49], in our experiment no reduction was detected in the chlorophyll concentration by the irrigation with different types of water (Figure 8). In general, no changes were observed in the chlorophyll fluorescence parameters studied, so the photochemical machinery was apparently not damaged by the phytotoxic elements in leaves of DSW-irrigated trees either. Therefore, the proportion of the absorbed energy being used in photochemistry (Φ_PSII_) was not modified by the DSW treatment (Figure 10). However, due to the reduction in the photosynthetic rate in July, the *A*/Φ_PSII_ ratio decreased in DSW-irrigated trees, indicating that reactive oxygen species (ROS) can be generated, since O_2_ can operate as an alternative acceptor, depleting the electron excess that is not utilized in metabolic processes [50]. Although oxidative stress was not evaluated in this study, previous results have shown that citrus trees have different capacities to overcome this oxidative stress generated by irrigation with DSW depending on the rootstock [20,21].

### 2.5. Plant Growth and Fruit Yield

The effect of irrigation with the different water sources on tree growth was evaluated throughout the experiment by conducting periodic measurements of the canopy (Figure 11). Our ‘Rio Red’ grapefruit was grafted onto a five-year-old *Citrus macrophylla* rootstock in June 2019, so the tree growth was very rapid. Appendix A displays the evolution of the overall size of the trees throughout the experiment. Trees grew exponentially during the first part of the experiment, producing an increase in the tree canopy from 0.1 m^3^ measured in November 2019 to almost 4 m^3^ one year later (Figure 11); after that, linear growth in the second half of the experiment increased the canopy volume from 4 to 8 m^3^. However, no significant changes in tree growth were observed with the treatments throughout the experiment, although a slightly higher growth of DSW–B-irrigated trees was observed from the beginning of 2021 onwards (Figure 11). Unfortunately, only limited information is available regarding the effect of irrigation with DSW in citrus trees on its agronomic behavior in the field. Under controlled conditions, young lemon trees irrigated with DSW had decreased shoot growth (mainly the leaf size) [20,21], whilst no effects were found on plant growth when similar water to that in our DSW–B treatment was tested [20].

Table 3 shows the effect of the irrigation with the different water sources on the fruit yield parameters obtained as the average yield achieved in 2021 and 2022. Due to the youth of the trees, no yield was obtained in 2020, and the yields of 2021 and 2022 were still low compared to commercial grapefruit yields. No significant effect of the irrigation with different water sources was observed in any of the fruit yield parameters studied. Previous results with similar treatments in a young orchard of mandarin trees showed no differences in the total yield or in the crop load during the first two years of the experiment [3]. It is therefore possible that differences may be observed as the tree production increases and stabilizes with maturity.

The high tree growth rate during the three-year study prevented a major accumulation of phytotoxic elements, by means of a dilution effect, avoiding toxic concentrations. It is likely that under a more stationary scenario where the trees have reached their maximum growth, leaves could accumulate phytotoxic elements in higher concentrations, which may alter the different nutritional, physiological, and metabolic processes, and ultimately affect tree growth and yield.

## 3. Materials and Methods

### 3.1. Experimental Plot, Vegetal Material, and Crop Management

The study was performed over three consecutive years (from June 2019 to December 2022) in a commercial farm located in Torre Pacheco, Murcia, Spain (37°47′30″ N; 1°03′85″ W; 30 m above sea level). The climate is semi-arid Mediterranean, featuring predominantly warm, arid summers and mild winters, with a mean annual rainfall of about 400 mm during the three years of the experiment, and a total annual reference evapotranspiration (ET_0_), calculated via the Penman–Monteith method, of about 1300 mm (three-year average).

The experiment was performed in a 0.28 ha ‘Rio Red’ grapefruit orchard (*Citrus* × *paradisi* Mac.) newly grafted on *Citrus macrophylla* Wester rootstock aged five years at the beginning of the experiment (June 2019), with a tree-spacing of 5.5 m × 3.5 m.

The irrigation system consisted of two polyethylene drip lines laid on the soil surface, one on either side of the tree. Four self-pressure compensating emitters per tree provided a discharge of 4 L h^−1^ each. The irrigation doses were scheduled based on the daily crop evapotranspiration accumulated during the previous week. No leaching fraction was added to the irrigation doses, regardless of water salinity. All the treatments received the same amounts of fertilizer (N-P_2_O_5_-K_2_O-CaO, MgO), supplied through the drip irrigation system. This enabled the possible effects of the incorporation of DSW to be analyzed; such effects could have been hidden if the fertilizer supply had been adjusted based on the nutrient content of the irrigation water. Pest control and pruning practices were those commonly used by growers in the area, and no weeds were allowed to develop within the orchard.

### 3.2. Treatments and Experimental Design

The treatments were established by supplying four water sources to the irrigation system: (i) fresh water (FW), provided by the Campo de Cartagena irrigators community; (ii) DSW provided by the coastal desalination plant of Escombreras (30 km from the farm); (iii) mixed water (MW), a 50% blend of FW and DSW (1:1); and (iv) DSW with a low B concentration (DSW–B). DSW from the Escombreras plant presented a B concentration of around 1.05 mg L^−1^, and an on-farm reverse osmosis system treated the DSW to reduce the B concentration (between 0.2 and 0.4 mg L^−1^) in the DSW–B treatment [23].

A total of four treatments were applied, corresponding to the four water resources previously described: FW, DSW, MW, and DSW–B. The lay-out of the experiment took the form of three completely randomized blocks with four experimental plots in each one (one treatment per plot). In each experimental plot (of 4 × 3 trees), border trees were excluded from the study to eliminate potential edge effects, and only the two central trees were studied.

### 3.3. Water Quality and Soil Analysis

Water samples were collected in glass bottles for each water source monthly throughout the experiment, transported in an icebox to the laboratory and stored at 5 °C until the physical and chemical analyses. The electrical conductivity (EC) was measured with a conductivity instrument GLP-31 (Crison Instruments S.A., Barcelona, Spain). Inductively coupled plasma (ICP-MS Agilent 7900, Agilent Technologies, Santa Clara, CA, USA) was used to determine the Na^+^, K^+^, and B concentrations. Chloride was quantified by ion chromatography with a liquid chromatograph (Thermo Scientific Dionex, Model ICS-2100, Thermo Scientific, Basel, Switzerland).

Three soil samples (one per block) for each irrigation treatment were collected every four months (in February, June, and October) in the area surrounding one central tree, at 0–0.50 m depths, and 0.30 m away from the emitter. Soluble salt contents were determined in the saturated paste extract [51]. Inductively coupled plasma (ICP-MS Agilent 7900) was used to determine the concentration of water soluble Na^+^. Extractable B was determined in soil samples by refluxing 20 g soil with 40 mL hot water (boiling) for a period of 5 min. One aliquot from the filtered extract was then used for measuring B using inductively coupled plasma (ICP-MS Agilent 7900).

### 3.4. Plant Mineral Analysis

Leaf samples were periodically taken for phytotoxic elements analysis. Old leaves were sampled during the first year of the experiment, with the oldest leaves on the tree being considered as the old leaves. On the other hand, the leaves that sprouted in spring (spring bud leaves) were also studied in 2020, 2021, and 2022. For each sampling, the two central trees of each experimental plot were sampled, giving a total of six samples per treatment (six trees per irrigation system). The leaf samples were rinsed in deionized water, freeze-dried, and ground for analytical determinations. In May 2021, fine roots of the 0–0.25 m layer were sampled in the same trees described above, they were carefully separated from the soil, rinsed with deionized water, oven-dried at 60 °C, and ground. The ground tissues (leaves and roots) were ashed, dissolved in 0.7 N HNO_3_, and Na^+^, K^+^, Ca^2+^, Mg^2+^, and B were determined by inductively coupled plasma optical emission spectrometry (Varian ICP-OES Vista MPX, Varian Inc., Palo Alto, USA). Chloride was extracted from 50 mg of ground plant material with 25 mL of deionized water and measured by ion chromatography with a liquid chromatograph (Model ICS-3000, Thermo Fisher Scientific Inc., Logan, UT, USA).

### 3.5. Plant Water Relations

Midday stem water potential (Ψ_stem_) was measured periodically in one mature, fully-expanded leaf in the middle third of the tree, in the same trees as the leaf mineral study (six trees per irrigation system). The leaves were enclosed within foil-covered plastic and aluminum envelopes at least two hours prior to the midday measurement [52]. In the same trees and on the same days as the Ψ_stem_ determination, the leaf water potential (Ψ_leaf_) was also measured in similar unwrapped leaves. Both Ψ_stem_ and Ψ_leaf_ were measured at noon (12:00–14:00), using a pressure chamber (model 3000; Soil Moisture Equipment Corp., Santa Barbara, CA, USA) and following the recommendations of Turner [53]. Following the Ψ_leaf_ measurement, the leaves were frozen and stored at −20 °C for determination of the leaf osmotic potential (Π) by a Wescor 5520 vapor pressure osmometer (Wescor, Logan, UT, USA). The leaf turgor potential was calculated as the difference between Ψ_leaf_ and Π.

### 3.6. Gas Exchange Parameters and Chlorophyll Fluorescence

Gas exchange measurements were taken periodically, coinciding with the plant water study, between 08:00 and 10:00 GMT, in daylight hours (to avoid high afternoon temperatures and air vapor pressure deficit). Measurements were made on healthy, fully expanded mature leaves (one leaf per tree, for the same six trees described above) exposed to the sun in exterior mid-canopy positions. The leaf net photosynthesis rate (*A*), stomatal conductance (*g_s_*), and leaf transpiration (*E*) were measured with a portable photosynthesis system (Li-6400, Li-Cor, Lincoln, NE, USA) equipped with a broad leaf chamber (6.0 cm^2^). The air flow rate inside the leaf chamber was 500 μmol s^−1^. The reference CO_2_ concentration was fixed at 400 μmol mol^−1^ through a CO_2_ injector system (6400-01 LiCOR, Lincoln, NE, USA). All measurements were made using a red–blue light source (6400-02B light emitting diode; Li-COR, Lincoln, NE, USA) and the PPFD was fixed at 1200 μmol m^−2^ s^−1^, which exceeds the saturating light intensity for photosynthesis of citrus leaves [54].

Chlorophyll fluorescence measurements were performed in parallel to those of the gas exchange parameters, on the same days and in the same leaves, using a pulse-modulated field-fluorescence monitoring system (FMS-2, Hansatech Instruments, Norfolk, UK). The chlorophyll fluorescence kinetics of leaves adapted to light were studied and the following parameters were measured: the antenna efficiency of PSII [F_v_′/F_m_′ = (F_m_′ − F_0_′)/F_m_′], the quantum efficiency of PSII [Φ_PSII_ = (F_m_′ − F_s_)/F_m_′], and the photochemical quenching coefficient [qP = (F_m_′ − F_s_)/(F_m_′ − F_0_′)], where F_v_’ is the variable fluorescence, F_m_′ is the maximum value when all reaction centers are closed after a pulse of saturating light (12,000 μmol m^−2^ s^−1^ for 0.8 s), F_0_′ is the minimal fluorescence in the light-adapted state that is obtained by turning off the actinic light temporarily and applying a pulse of far-red light (735 nm) to drain the electrons from PSII, and F_s_ is the steady-state fluorescence yield.

### 3.7. Osmolytes and Chlorophyll Determinations

The leaf samples for mineral analysis were used for osmolytes and chlorophyll analysis. Proline was extracted from 50 mg of leaf tissue with sulfosalicylic acid (3%) and quantified according to the protocol described by Bates et al. [55]. Quaternary ammonium compounds (QAC) were extracted from dry tissue with 1 M H_2_SO_4_ and quantified using a glycine betaine standard curve, according to the method described by Grieve and Grattan [56].

Chlorophyll contents were estimated using the procedure described by Inskeep and Blomm [57], extracting 20 mg of ground material with N,N-dimethylformamide and measuring the absorbance at 664.5 and 647 nm in a Shimadzu UV-1800 spectrophotometer (Shimadzu Corporation, Kyoto, Japan).

### 3.8. Vegetative Growth and Yield Fruit

Measurements were made in the two central trees of each experimental plot. The trunk circumference of the rootstock, tree canopy height, and perimeter were periodically measured during the experimental period in the same six trees per treatment. Canopy volume (V) was calculated considering the canopy diameter (D) and height (H), according to the equation of Turrell [58]: V = 0.5238 H D^2^, where V, D, and H are in m^3^, m, and m, respectively.

In early February 2022 and late December 2022 (2021 and 2022 yields, respectively), the individual tree yield was measured for six trees per treatment (two trees per experimental plot. The number of fruits and the total fruit weight of each tree were measured.

### 3.9. Statistical Analysis

All data were analyzed using the analysis of variance (ANOVA) procedures in Statgraphics Plus 5.1 software (Statistical Graphics Corporation, Warrenton, VA, USA). When there was a significant effect (value of *p* < 0.05), means were separated using Duncan’s multiple range test. The Pearson correlation coefficients between the concentrations of phytotoxic elements in water, soil, and leaves were calculated using the same statistical software.

## 4. Conclusions

The data obtained in this study highlight the interaction between irrigation water quality and plant behavior. While the investigation revealed varying concentrations of phytotoxic elements in the irrigation waters used for citrus, with notably elevated levels of boron, the rapid growth of the trees seemed to mitigate potential adverse effects due to a dilution effect. Despite boron concentrations nearing toxicity thresholds in leaf tissues, the overall impact on tree growth and fruit yield was not significant.

This research has practical implications for agricultural management, particularly in semi-arid regions reliant on alternative water sources like desalinated seawater (DSW). The findings suggest that while DSW irrigation may introduce elevated levels of certain phytotoxic elements, the rapid growth rates can buffer against harmful effects, at least in the short to medium term.

However, the study also highlights the need for continued investigation, especially over extended periods, to fully elucidate the potential long-term impacts of DSW irrigation on citrus cultivation. Future research should delve deeper into understanding how prolonged exposure to elevated phytotoxic element concentrations might affect nutritional, physiological, and metabolic processes within the citrus trees. Moreover, examining the implications of DSW irrigation under more stable growth conditions, where trees have reached maturity, could provide valuable insights into potential risks.

In summary, while the initial hypothesis regarding the alteration of physiological responses due to DSW irrigation was not fully supported by this study, the applied nature of the research underscores the importance of ongoing research. Longer-term investigations will be essential to gain more knowledge of the relationship between water quality, plant behavior, and agricultural productivity, ultimately guiding sustainable irrigation practices in semi-arid environments.

## Figures and Tables

**Figure 2 plants-13-00781-f002:**
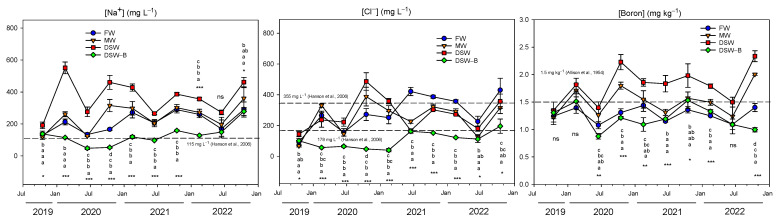
Evolution of the concentration of water-soluble Na^+^ and Cl^−^ and the extractable B in soil samples collected at the 0–0.50 m depth and at 0.30 m from the emitter. The dashed line in the figure determines the phytotoxicity threshold suggested for Na^+^ [29], Cl^−^ [29], and B [33]. Initials represent the four irrigation treatments (FW: fresh water; MW: mixed water; DSW: desalinated seawater; and DSW–B: DSW with reduced boron). Data were recorded between October 2019 and December 2022. Each point represents the average of six samples. * *p* < 0.05; ** *p* < 0.01; *** *p* < 0.001; ns: not significant. For each date, different letters indicate significant differences according to Duncan’s multiple range test at the 95% confidence level.

**Figure 3 plants-13-00781-f003:**
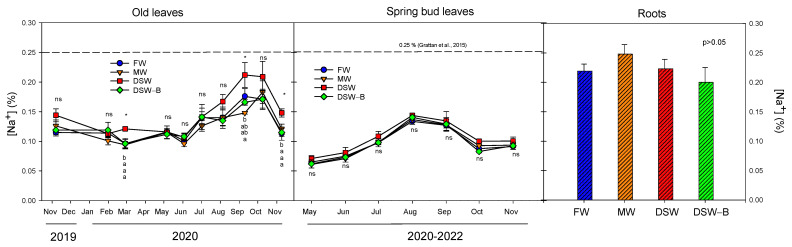
Effect of the irrigation water on the evolution of the concentration of Na^+^ in old leaves (2019–2020), spring bud leaves (average of 2020–2022), and roots (May 2021). The dashed line in the figure determines the phytotoxicity threshold suggested for Na^+^ [26]. Initials represent the four irrigation treatments (FW: fresh water; MW: mixed water; DSW: desalinated seawater; and DSW–B: DSW with reduced boron). * *p* < 0.05; ns: not significant. For each date, different letters indicate significant differences according to Duncan’s multiple range test at the 95% confidence level.

**Figure 4 plants-13-00781-f004:**
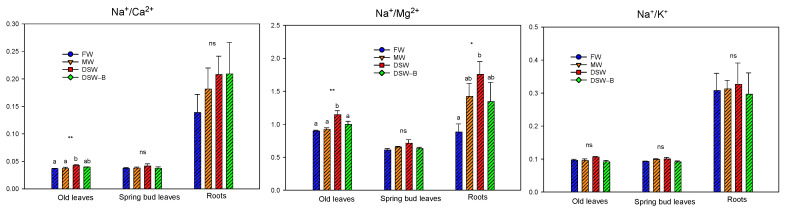
Effects of the irrigation water on the Na^+^/Ca^2+^, Na^+^/Mg^2+^, and Na^+^/K^+^ ratios in old and spring bud leaves, and in roots. Initials represent the four irrigation treatments (FW: fresh water; MW: mixed water; DSW: desalinated seawater; and DSW–B: DSW with reduced boron). Data were averaged from samples taken between November 2019 and November 2020 for old leaves, and samples taken between June 2020 and December 2022 for spring bud leaves. Roots were sampled in May 2021. * *p* < 0.05; ** *p* < 0.01; ns: not significant. In each tissue, different letters indicate significant differences according to Duncan’s multiple range test at the 95% confidence level.

**Figure 5 plants-13-00781-f005:**
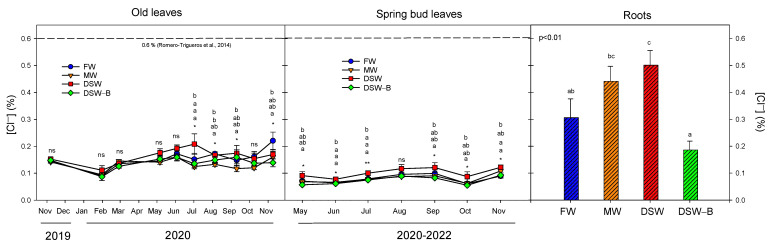
Effect of the irrigation water on the evolution of the concentration of Cl^−^ in old leaves (2019–2020), spring bud leaves (average of 2020–2022), and roots (May 2021). The dashed line in the figure determines the phytotoxicity threshold suggested for Cl^−^ [38]. Initials represent the four irrigation treatments (FW: fresh water; MW: mixed water; DSW: desalinated seawater; and DSW–B: DSW with reduced boron). * *p* < 0.05; ** *p* < 0.01; ns: not significant. For each date, different letters indicate significant differences according to Duncan’s multiple range test at the 95% confidence level.

**Figure 6 plants-13-00781-f006:**
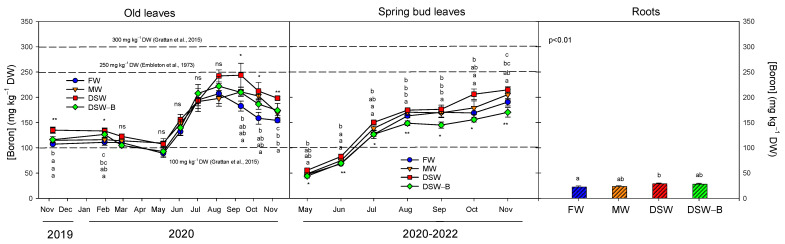
Effect of the irrigation water on the evolution of the concentration of boron in old leaves (2019–2020), spring bud leaves (average of 2020–2022), and roots (May 2021). The dashed line in the figure determines the phytotoxicity threshold suggested for B [26,40]. Initials represent the four irrigation treatments (FW: fresh water; MW: mixed water; DSW: desalinated seawater; and DSW–B: DSW with reduced boron). * *p* < 0.05; ** *p* < 0.01; ns: not significant. For each date, different letters indicate significant differences according to Duncan’s multiple range test at the 95% confidence level.

**Figure 7 plants-13-00781-f007:**
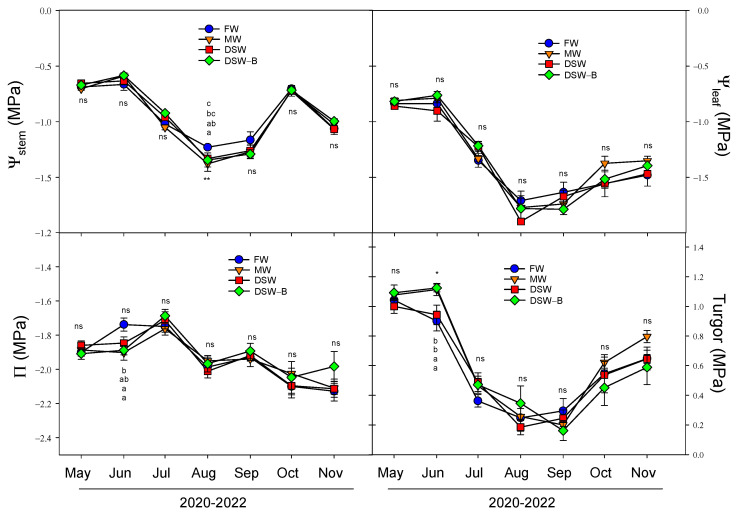
Effect of the irrigation water on the evolution of the midday stem water potential (Ψ_stem_), leaf water potential (Ψ_leaf_), osmotic potential (Π), and leaf turgor in spring bud leaves (average of 2020, 2021, and 2022) throughout the experiment. Initials represent the four irrigation treatments (FW: fresh water; MW: mixed water; DSW: desalinated seawater; and DSW–B: DSW with reduced boron). * *p* < 0.05; ** *p* < 0.01; ns: not significant. For each date, different letters indicate significant differences according to Duncan’s multiple range test at the 95% confidence level.

**Figure 8 plants-13-00781-f008:**
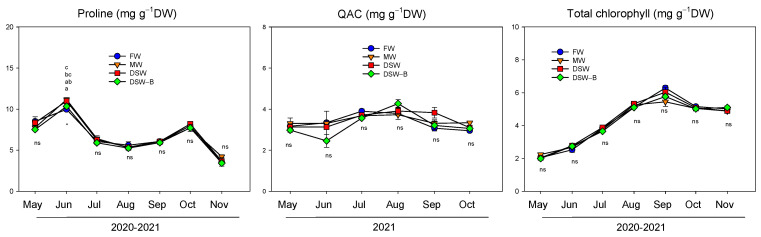
Effect of the irrigation water on the evolution of the proline, quaternary ammonium compounds (QACs), and total chlorophyll concentration in mature leaves (average of 2020, 2021 for proline and chlorophyll, and 2021 for QACs). Initials represent the four irrigation treatments (FW: fresh water; MW: mixed water; DSW: desalinated seawater; and DSW–B: DSW with reduced boron). * *p* < 0.05; ns: not significant. For each date, different letters indicate significant differences according to Duncan’s multiple range test at the 95% confidence level.

**Figure 9 plants-13-00781-f009:**
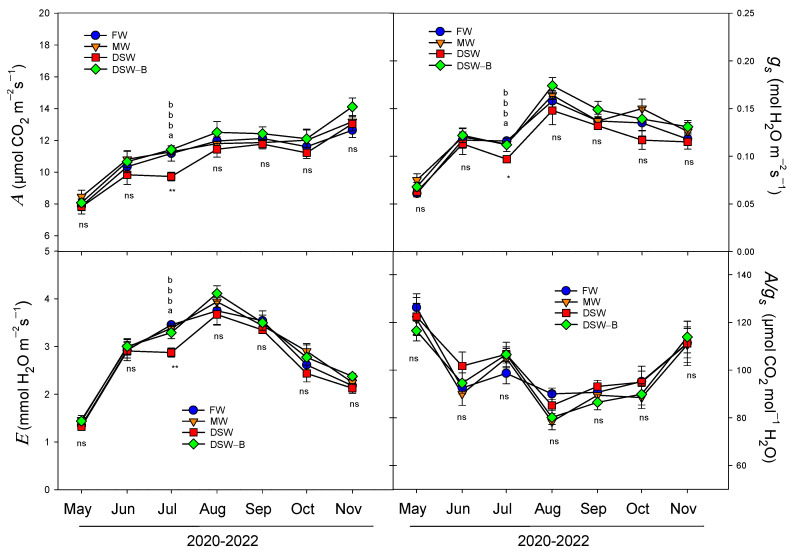
Effect of the irrigation water on the evolution of net photosynthesis (A), stomatal conductance (g_s_), transpiration rate (E), and intrinsic water use efficiency (A/g_s_) in mature leaves (average of 2020, 2021, and 2022) throughout the experiment. Initials represent the four irrigation treatments (FW: fresh water; MW: mixed water; DSW: desalinated seawater; and DSW–B: DSW with reduced boron). * *p* < 0.05; ** *p* < 0.01; ns: not significant. For each date, different letters indicate significant differences according to Duncan’s multiple range test at the 95% confidence level.

**Figure 10 plants-13-00781-f010:**
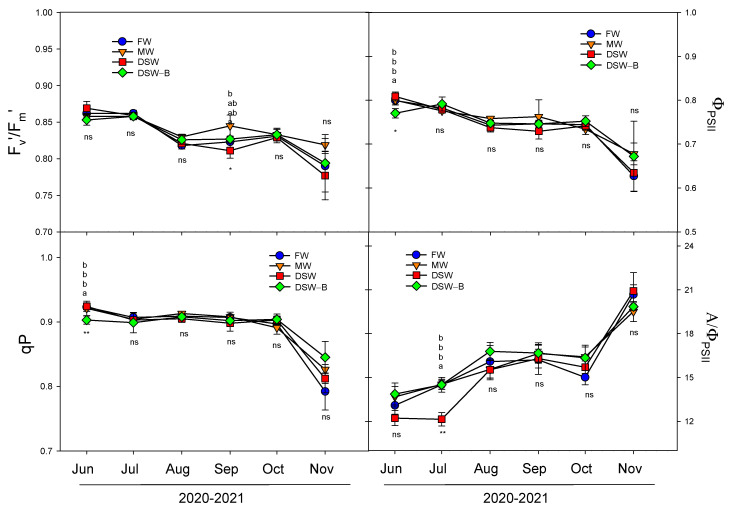
Effect of the irrigation water on the evolution of the efficiency of the antennas from PSII (F’_v_/F’_m_), photochemical efficiency of PSII (Φ_PSII_), photochemical quenching (qP), and A/Φ_PSII_ ratio in spring bud leaves (average of 2020 and 2021) throughout the experiment. Initials represent the four irrigation treatments (FW: fresh water; MW: mixed water; DSW: desalinated seawater; and DSW–B: DSW with reduced boron). * *p* < 0.05; ** *p* < 0.01; ns: not significant. For each date, different letters indicate significant differences according to Duncan’s multiple range test at the 95% confidence level.

**Figure 11 plants-13-00781-f011:**
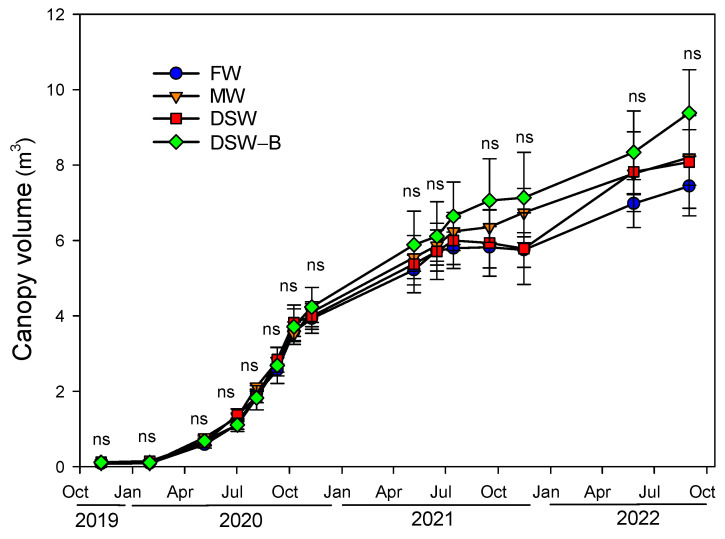
Effect of the irrigation water on the evolution of the canopy volume of the trees throughout the experiment (from November 2019 to December 2022). Initials represent the four irrigation treatments (FW: fresh water; MW: mixed water; DSW: desalinated seawater; and DSW–B: DSW with reduced boron). ns: not significant at the 95% confidence level.

**Table 1 plants-13-00781-t001:** The electrical conductivity (EC, dS m^−1^) and average ionic composition (mg L^−1^) of the water sources used during the experimental work (FW: fresh water; MW: mixed water; DSW: desalinated seawater; and DSW–B: DSW with reduced boron). Data were averaged from 40 samples taken between June 2019 and December 2022.

Water Source	EC	Ca^2+^	Mg^2+^	K^+^	NO_3_^−^	PO_4_^3−^	SO_4_^2−^
FW	1.23 d	71.4 d	42.7 d	5.88 b	4.92 b	0.88 bc	209.2 c
MW	1.00 c	45.8 c	20.4 c	6.22 b	2.57 c	1.15 c	90.6 b
DSW	0.89 b	24.5 b	5.3 b	6.89 b	1.31 d	0.36 ab	7.4 a
DSW–B	0.17 a	2.7 a	1.0 a	0.41 a	0.78 a	0.16 a	1.3 a
ANOVA	***	***	***	***	***	**	***

** *p* < 0.01; *** *p* < 0.001; ns: not significant. Different letters indicate significant differences according to Duncan’s multiple range test at the 95% confidence level.

**Table 2 plants-13-00781-t002:** Matrix of Pearson’s correlation coefficients obtained between the concentrations of Na^+^, Cl^−^, and B found in the spring bud leaves. Data were collected three times per year (February, June, and October) from October 2019 to October 2022.

	[Na^+^]_water_	[Na^+^]_soil_	[Na^+^]_leaf_	[Cl^−^]_water_	[Cl^−^]_soil_	[Cl^−^]_leaf_	[B]_water_	[B]_soil_	[B]_leaf_
[Na^+^]_water_	-	**0.525 *****	0.180	**0.982 *****	**0.547 *****	0.187	**0.661 *****	**0.365 ***	0.129
[Na^+^]_soil_	**0.525 *****	-	0.107	**0.517 ****	**0.725 *****	0.021	**0.720 *****	**0.812 ****	**0.521 ****
[Na^+^]_leaf_	0.180	0.107	-	0.234	0.0756	0.203	0.236	0.245	**0.529 *****
[Cl^−^]_water_	**0.982 *****	**0.517 ****	0.234	-	**0.526 *****	0.204	**0.691 *****	**0.380 ***	0.113
[Cl^−^]_soil_	**0.547 *****	**0.725 *****	0.076	**0.526 *****	-	−0.244	**0.425 ****	**0.588 *****	**0.439 ****
[Cl^−^]_leaf_	0.187	0.021	0.203	0.204	−0.244	-	0.293	0.024	0.135
[B]_water_	**0.661 *****	**0.720 *****	0.236	**0.691 *****	**0.420 ****	0.293	-	**0.725 *****	0.302
[B]_soil_	**0.365 ***	**0.812 ****	0.245	**0.380 ***	**0.588 *****	0.024	**0.725 *****	-	**0.570 *****
[B]_leaf_	0.129	**0.521 ****	**0.529 *****	0.113	**0.439 ****	0.135	0.302	**0.570 *****	-

The numbers in bold indicate significant differences. *, **, and *** indicate significant differences at the 0.05, 0.01, and 0.001 levels of probability, respectively.

**Table 3 plants-13-00781-t003:** Effect of the irrigation water on the fruit yield parameters during the experimental period for each irrigation treatment (FW: fresh water; MW: mixed water; DSW: desalinated seawater; and DSW–B: DSW with reduced boron). The data are averages of 2021 and 2022.

Water Source	Yield (kg tree^−1^)	Fruit Number	Fruit Weight (g)
FW	85.1 ± 4.2	229.5 ± 16.1	380.5 ± 13.7
MW	81.0 ± 6.2	222.0 ± 15.4	367.6 ± 5.9
DSW	91.5 ± 4.7	246.5 ± 16.7	382.8 ± 22.8
DSW–B	90.4 ± 6.6	248.4 ± 18.4	368.8 ± 22.8
ANOVA	ns	ns	ns

The values after the ± sign represent the standard deviation of the data. ns: not significant at 95% confidence level.

## Data Availability

Data are contained within the article and Appendix A.

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
