# Peer review of "Physiological Responses of a Grapefruit Orchard to Irrigation with Desalinated Seawater"

_plants, 2024, doi:10.3390/plants13060781_

Round 1

Reviewer 1 Report

Comments and Suggestions for Authors

The manuscript by Navarro et al. entitled "Physiological Responses of a Grapefruit Orchard to Irrigation with Desalinated Seawater" aims to initiate a long-term experiment to provide guidance on the effects of DSW utilization in semi-arid areas. This is an interesting and meaningful study. I agree that this paper will be accepted for publication, but revisions are required before doing so.

Main comments

1. In this study, how do grapefruits grow after being irrigated with different water sources? Can the authors provide some pictures, including the growth of the whole plant or the shape of some leaves, such as old leaves, new leaves, and buds?

What are the effects of phytotoxic elements (Na+, Cl, and B) on grapefruit growth, including root and leaf development? Photosynthesis-related data is still not intuitive enough. 

2. Concentrations of Na+ and Cl exceeded recommended thresholds for citrus (except in the DSWB treatment), and B concentrations in DSW and MW also exceeded established toxicity thresholds for citrus. Irrigation with this water for three years resulted in the accumulation of these elements mainly in the soil rather than in the aboveground parts. This also makes me curious, was there any change in the root development of grapefruit compared to normal irrigation water?  

3. In Figs 3-5 (roots) and Fig 4, just the variance value of the data doesn't seem to be a good way to show the differences between treatments.

Author Response

Thank you very much for your comments and professional advice. These opinions help to improve academic rigor of our manuscript. Based on your suggestion and request, we have made various modifications on the revised manuscript.

Main comments

1. Thank you for your interest in the study. We have included supplementary material to this paper, in which some images of the trees and leaves are included: the growth of the threes throughout the experiment, the final size of the trees, and the shape of buds, young and old leaves. Since there were no differences between treatments (toxic effects on the leaves), only one image of each type of leaf from the DSW irrigation treatment was added.

In this experiment we have measured the plant growth throught the canopy volume study but no study of root growth has been made. From the beginning of the experiment, trees experienced a very fast growth, from 0.1 m3 in November 2019 to 8 m3 in November 2022 (some pictures ilustrating this growth has been included in the Supplementary Material). The rapid growth of the trees over the three-year investigation period hindered significant buildup of Na+, Cl, and B (Figures 3, 5 and 6), thanks to a dilution process, thereby preventing hazardous levels of concentration. We hypotised that, due to this dilution effect, important physiological and metabolic processes (as the photosynthesis rate), were not affected by the accumulation of phytotoxic elements.

2. The reviewer is right, the Na+ and Cland B concentrations in most of the treatments used in the experiment were higher than the thresholds established as toxic for citrus. This resulted in accumulation in the soil rather than leaves due to the dilution effect explained in the Response 2. Due to these high concentrations of phytotoxic elements in the soil around the roots, one might think that root growth was affected. Unfortunately, root growth measurements were not taken during the experiment, so we do not have data on root development under these conditions. However, based on data of phytotoxic elements concentrations in Figures 3, 4 and 6, it could be inferred that the roots were also growing sufficiently to prevent significant accumulation of most phytotoxic elements from accumulating (only Cl- was accumulated in the roots of trees irrigated with DSW).

3. Roots were sampled in May 2021 and analyzed to study their mineral composition. The concentrations of Na+, Cland B are shown in Figures 3, 5 and 6 respectively. Figure 4 also displays the Na+/Ca2+, Na+/Mg2+, and Na+/K+ ratios in roots. Each bar in these figures represents the average of six replicates (the six trees per treatment described in the Material and Methods section). The analysis of variance and the separation of means when ANOVA was significant (value of p<0.05) were performed similarly to the rest of the data presented in this study. Unlike the leaves, the analysis of the roots involved a single sampling. Only one sampling was carried out to preserve the root system of the tree as much as possible and avoid damaging it with periodic sampling. However, we think that the results obtained provide valuable information about the absorption and accumulation of elements in the roots.

Reviewer 2 Report

Comments and Suggestions for Authors

I have had the pleasure of reviewing the manuscript titled "Physiological Responses of a Grapefruit Orchard to Irrigation with Desalinated Seawater"  by Josefa Maria Navarro, Alberto Imbernón-Mulero, Juan Miguel Robles, Francisco Miguel Hernández Ballester, Vera Antolinos, Belen Gallego-Elvira, and José F. Maestre-Valero. The study explores the short-term impacts of different irrigation waters, including desalinated seawater (DSW), on a 'Rio Red’ grapefruit orchard.

The research addresses a critical issue - water scarcity in agriculture - and evaluates the potential of desalinated seawater as an irrigation solution. The three-year experiment offers valuable insights into the effects of various irrigation waters on citrus trees, with a focus on phytotoxic elements such as sodium, chloride, and boron.

The authors successfully highlight the challenges associated with adopting desalinated seawater, including the presence of harmful elements. The experiments involving freshwater (FW), DSW, a mix of FW and DSW (MW), and DSW with low boron concentration (DSW−B) provide a comprehensive understanding of the potential risks and benefits associated with each irrigation water type.

I appreciate the meticulous documentation of concentrations of phytotoxic elements in irrigation waters and their subsequent impact on citrus trees. The finding that sodium and chloride concentrations exceeded citrus thresholds in all treatments, except in DSW−B, is particularly noteworthy. The discussion on leaf concentrations and their implications on plant nutrition and physiology, including gas exchange and chlorophyll levels, adds depth to the study.

The authors effectively communicate that the rapid growth of the trees acts as a protective mechanism against the accumulation of phytotoxic elements, preventing significant impacts on physiological processes. The insight that the dilution effect plays a crucial role in mitigating potential harm to the grapefruit orchard is an important contribution to the field.

The manuscript concludes with a thoughtful consideration of the future implications, emphasizing the need for further study until the trees reach maturity. This forward-looking approach acknowledges that the long-term effects of desalinated seawater irrigation may vary as trees mature.

I only have a few minor comments on this manuscript. I list them below:

1) Please list keywords in alphabetical order.

2) Introduction - section lines 88-104 presents the research objectives in great detail, but it is also necessary to clearly outline the research hypothesis and the application significance of this research.

3) Lines 128 and 187 - no italic font is needed here.

4) Tables 1 and 3 - please explain the meaning of the abbreviations FW, MW, DSW and DSW-B under the table. The same applies to Figures 1-11. Please remember that all abbreviations used in Figures and Tables must be explained above or below them.

5) I find Figure 4 incomplete and incomprehensible to me.

6) Table 3 - please provide information under the table what the value after the +/- sign means.

7) Conclusions - I don't like this section. I believe it needs to be thoroughly reworded. This is mainly a comprehensive summary of the results. In this chapter, you should answer the question whether the research objectives were met. You should write whether the research hypothesis has been proven true. It is necessary to emphasize the applied nature of the research and outline the direction of further research.

In summary, the manuscript provides valuable information for researchers, policymakers, and practitioners involved in agricultural water management. The study's well-designed experiments, thorough analysis, and clear presentation make it a significant contribution to the scientific community. I recommend acceptance of this manuscript for publication in Plants, as it addresses an important and timely issue with scientific rigor and relevance.

Author Response

Thank you very much for your comments and professional advice. These opinions help to improve academic rigor of our manuscript. Based on your suggestion and request, we have made various modifications on the revised manuscript.

Main comments:

1. Thank you for your comment. We have corrected the list of keywords in alphabetic order.

2. We agree with the reviewer. The research hypothesis and the application significance of the research are missing. We have added some sentences to incorporate them.

3. Thank you for the advice. According to the suggestion of the reviewer, we have changed the italic font to regular in lines 128, 187 and 476.

4. According to the suggestion of the reviewer, we have added the meaning of the abbreviations FW, MW, DSW and DSW-B in Tables 1 and 3 and Figures 1-11.

5. This figure represents the Na+/Ca2+, Na+/Mg2+, and Na+/K+ ratios in differents organs of the plant: old leaves, spring bud leaves, and roots. All these organs were sampled in different periods: old leaves from November 2019 to November 2020, spring bud leaves from June 2020 to December 2022, and roots in May 2021. We have revised this figure and the reviewer was right, information was missing from the figure description. We have incorporated it to clarify the figure.

6. We have added the meaning of the values after the +/- sign.

7. According to the suggestions of the reviewer, the Conclusions section has be completely modified, responding if the objectives have been achieved, if the hypothesis has been tested, and emphasizing the applied nature of the research and the direction of future research.

8. Thank you very much for your comments.
